# Improving fishing ground estimation with weak supervision and meta-learning

**Kazuki Takasan**⊕, **Masaaki Iiyama**⊕⊕*

Graduate School of Data Science, Shiga University, Hikone, Shiga, Japan

⊕ These authors contributed equally to this work.
* iiyama@iiyama-lab.org

**Data availability statement:** Fishing vessel trajectory data are publicly available from Global Fishing Watch, https://globalfishingwatch.org/datasets-and-code/. Fish catch data cannot be shared publicly because of owned a third party. The data are owned by Shizuoka Prefectural Research Institute of Fishery and Ocean (https://fish-exp.pref.shizuoka.jp/). Researchers who meet the criteria for access to confidential data would be able to access these data.

## Abstract

Estimating fishing grounds is an important task in the fishing industry. This study modeled the fisher's decision-making process based on sea surface temperature patterns as a pattern recognition task. We used a deep learning-based keypoint detector to estimate fishing ground locations from these patterns. However, training the model required catch data for annotation, the amount of which was limited. To address this, we proposed a training strategy that combines weak supervision and meta-learning to estimate fishing grounds. Weak supervision involves using partially annotated or noisy data, where the labels are incomplete or imprecise. In our case, catch data cover only a subset of fishing grounds, and trajectory data, which are readily available and larger in volume than catch data, provide imprecise representations of fishing grounds. Meta-learning helps the model adapt to the noise by refining its learning rate during training. Our approach involved pre-training with trajectory data and fine-tuning with catch data, with a meta-learner further mitigating label noise during pre-training. Experimental results showed that our method improved the F1-score by 64% compared to the baseline using only catch data, demonstrating the effectiveness of pre-training and meta-learning.

## Introduction

Estimating fishing grounds, the areas where fish are most likely to be found and caught, is an important task in the fishing industry. The accurate estimation enhances the efficiency of fishing activities by guiding fishers to areas with higher catch potential, thus optimizing resources such as fuel and reducing the operational costs. Traditionally, fishers rely on empirical methods considering sea environmental information such as sea surface temperature (SST). However, the accuracy of these methods remains inconsistent due to the complexity of marine environments and variations in the fishers' skill.

Previous studies have employed habitat suitability index (HSI) models [1–4] and statistical approaches such as generalized additive models (GAM) and maximum entropy models [5–10] to understand the relationship between sea environmental variables and fishing grounds. These methods typically focus on point-wise environmental variables, which fail to capture the broader, regional cues fishers use in their decision-making processes.

**Funding:** Japan Society for Promotion of Science, KAKENHI, Grant Number 21H04913. Japan Science and Technology Agency, CREST, Grant Number JPMJCR19F1. The funders had no role in study design, data collection and analysis, decision to publish, or preparation of the manuscript.

**Competing interests:** The authors have declared that no competing interests exist.

Recent advances in pattern recognition have inspired novel approaches that treat fishing ground estimation as a pattern recognition task. Our previous work [11,12] demonstrated the effectiveness of keypoint detection methods for fishing ground estimation by detecting regional SST patterns. These approaches emulate the process by which fishers implicitly read the two-dimensional pattern from the SST map to determine the fishing grounds. This can be defined as a general image recognition task, and deep learning-based methods have shown high performance. However, a significant challenge remains due to the lack of reliable annotated data for model training. Due to the scarcity of accurate catch data, the models cannot be adequately trained, resulting in reduced prediction accuracy.

In this paper, we propose a novel approach to tackle this challenge through weakly supervised pre-training and meta-learning. To compensate for the scarcity of precise catch data, we use fishing vessel trajectory data as weak supervision. While trajectory data is less accurate as an indicator of fishing grounds, it is widely available and provides useful patterns related to fishing activity. In the pre-training phase, the model learns these broader patterns from the trajectory data, which helps it learn characteristics of SST patterns. This process strengthens the model's performance, allowing it to perform better when fine-tuned on limited but more accurate catch data. This approach is based on our previous preliminary study [13], and the current study aims to refine and enhance the methodology.

Additionally, to mitigate the effects of noisy labels that exist in trajectory data, we introduce meta-learning. Trajectory data may include locations where vessels visited but did not engage in fishing activities, made no catch, or areas unsuitable for the target fish species, leading to noisy labels that do not accurately represent fishing grounds. Inspired by previous work [14], we employ a meta-learner to help the model adapt to these unreliable labels during pre-training. The meta-learner assesses label reliability, ensuring that noisy or misleading data has less impact on the model's learning process. While our study is consistent with their settings, focusing on many unreliable labels and a few reliable ones, our approach differs in that we address a detection task rather than the classification task they target. This distinction requires an adapted meta-learning strategy for effective implementation. As a result, our model exhibits improved robustness and enhanced performance when fine-tuned with more accurate catch data.

Our main contributions are summarized as follows.

- Our proposed method offers the advantages of addressing data scarcity using trajectory data during pre-training, effectively overcoming the scarcity of annotated data, and making the model more adaptable to real-world scenarios where available catch records may be limited.
- Furthermore, the integration of meta-learning enhances the model's robustness by reducing the impact of noisy labels during pre-training, ensuring improved performance in the face of uncertainties associated with weakly supervised data.
- A new application of weakly supervised learning and meta-learning in keypoint detection is presented, allowing for novel approaches to insufficient data sources prevalent in various domains.

## Related work

### Fishing ground estimation

**Ecological and statistical approach**  Since the development of remote-sensing technology, several studies have attempted to estimate the location of fishing grounds using ocean environmental conditions such as SST and sea surface height (SSH).

Studies with an ecological approach [1–4] employ HSI model, which is a conceptual framework that explains the relationship between environmental factors and the distribution of a given species. It quantifies the suitability of an environment for a species at a particular location, scaling from 0 to 1, where this value is referred to as the HSI.

In the statistical approaches [5–10], GAMs or maximum entropy models are used to solve a regression problem with catch per unit effort (CPUE) as the objective variable. In recent years, machine learning models such as support vector machine (SVM), random forest, and neural networks have been used and have demonstrated improved performance in this task [15–19].

Our study differs from most previous studies in predicting daily fishing grounds. In contrast, most previous studies have only predicted fishing grounds on a seasonal or monthly basis. Predicting daily changes in fishing grounds requires a large amount of data, which is difficult to collect. Therefore many studies have focused on seasonal predictions (which can be analyzed with less data). In contrast, we propose a method compensating for the lack of data by using weakly supervised learning to make daily predictions.

**Pattern recognition approach**   Traditional methods focus on point-wise environmental variables; fishermen often rely on broader, region-wise environmental cues from their surrounding area, such as eddies and tidal patterns, to find fishing grounds.

In our previous work [20], we used a two-dimensional SST map as an input for predicting fishing grounds using SVM and spectral clustering. Furthermore, in [11–13], fishing ground estimation was based on image recognition techniques, including object and keypoint-detection models, using 2D sea environmental data such as SST. While these approaches can incorporate information of data over a larger area, they face the challenge of partial annotation, where only some of the ground-truth points in the image are labeled. We addressed this issue by creating a specialized loss function [12] and expanding the supervised data using fishing vessel trajectories [13]. In this study, we extend trajectory data using it as weakly supervised data for model pre-training.

**Application of fishing vessel trajectories**   Fishing vessel trajectory data have identified fishing activities, including detecting of illegal fishing [21–23]. In addition, researchers have also used these trajectories to estimate the location of fishing grounds. Previous studies [9,18, 19,24] classified locations as fishing grounds or non-fishing grounds based on the speed of vessels and subsequently labeled these areas accordingly. In this study, we employ a similar approach to extract potential fishing ground locations from the trajectory data, using this data as weak supervision. Although trajectory data is inherently noisy, it serves as a large-scale source of information about fishing activities. A similar strategy has been successfully applied in other domains, such as vision-language representation learning, where noisy data (e.g., image-alt text pairs) was used to train models with impressive results despite the noise [42]. In our case, despite the noise in the trajectory data, its large-scale nature makes it a valuable signal for learning fishing ground locations.

## Keypoint detection

Keypoint detection is the task of detecting characteristic points in an image. A typical application of this task is human pose estimation, addressing the detection of human joints in an image and estimating human posture. Since the development of deep learning techniques, methods using deep neural networks (DNN) or convolutional neural networks (CNN) have been proposed [25–31].

There are two approaches for estimating keypoints: direct regression of keypoint coordinates [25–27] and heatmap regression [28–31]. In the latter approach, a 2D Gaussian kernel is

applied to the ground-truth coordinates to create a ground-truth heatmap and minimize the loss to the output heatmap. In addition to being easier to implement and more accurate [32], the heatmap regression is more suitable for fishing ground estimation, given that the output heatmaps are used as recommendation maps for fishers. Therefore, we set fishing grounds as keypoints, estimate the locations from 2D SST patterns as input images, and output the estimated locations as heatmaps. We do not consider the connection between the keypoints, and only use the output heatmap for the estimation.

## Weakly supervised learning

Weakly supervised learning uses incomplete, inexact, and inaccurate supervised data [33]. This approach includes situations where only a subset of the dataset is labeled (also referred to as semi-supervised learning [34]), situations with inaccurate labels (referred to as noisy labels [35]), and situations with coarse-grained labels, such as assigning labels to entire images rather than individual objects within the images [36–38]. Partially annotated data, characterized by missing labels for some objects in images, is a form of incomplete supervised data and is classified as weakly supervised learning. Researchers have addressed this challenge by adapting the loss function [39], employing specific sampling techniques [40], and self-supervised techniques [41]. The annotated data used in our study is considered partially annotated, either without noise or including noisy labels, since catch data do not cover all potential fishing grounds, and the representation of fishing ground locations through trajectory data is inaccurate.

In situations where there are numerous weakly labeled data and few strongly labeled data, a typical approach involves initial pre-training with weak labels followed by fine-tuning with strong labels [42–45]. This strategy is commonly seen in transfer learning, where models are pre-trained on a large, often general-purpose dataset, and then fine-tuned on a smaller, domain-specific dataset to adapt the model to the specific task. However, in our case, the available labeled data are weak and imprecise, and we lack access to sufficiently large pre-training datasets typical in transfer learning, making weakly supervised learning more suitable. However, noisy labels during pre-training can negatively affect model's performance. We introduced meta-learning to address this issue and adjust the influence of noisy labels on weakly annotated data during the training [14]. Their work follows the approach proposed by Andrychowicz et al. [46], expanding the application of meta-learning to weakly supervised learning.

## Methods

### Keypoint detection as fishing ground estimation

In this research, we addressed the challenge of accurately determining the locations of productive fishing grounds using SST maps, framing this task as a keypoint detection. This approach is inspired by the estimation process of fishers, who use 2D patterns of SST such as eddies and tides to find good fishing grounds. Like human pose estimation, a classic example of keypoint detection, we treated fishing grounds as keypoints in our fishing ground estimation and estimated their locations. The input for this task is a 2D image corresponding to a two-dimensional representation of SST, as Fig 1 shows.

Instead of directly predicting the coordinates of the keypoints, our model outputs a prediction heatmap, as Fig 2 shows. Areas with higher values on this heatmap indicated a higher likelihood of being fishing grounds and are thus suggested to fishermen as potential fishing

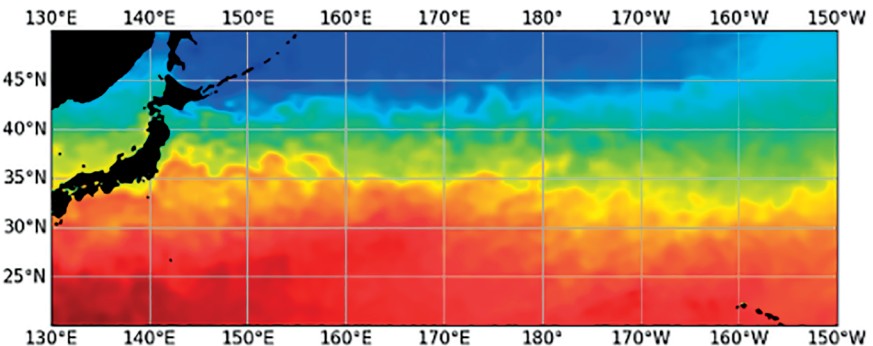

**Fig 1. SST map as an input image.** The coloring is for visualization; the image used for training is a two-dimensional array containing the SST itself.

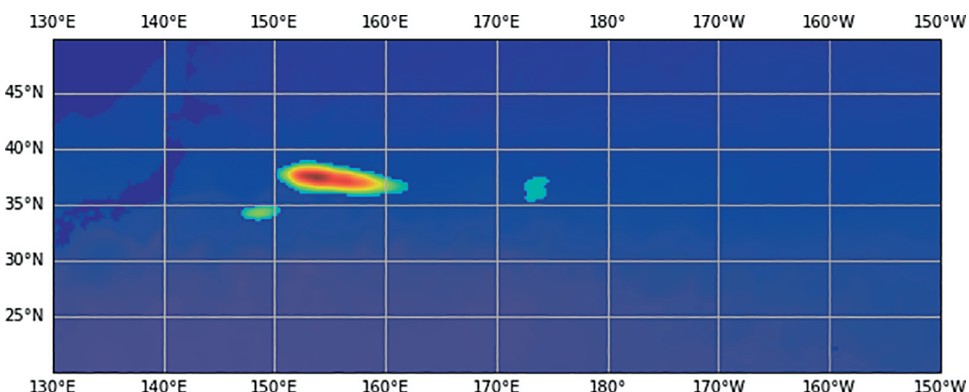

**Fig 2. Output heatmap.** The areas with higher values in the heatmap indicate a higher likelihood of fishing grounds.

locations. The model' training requires a ground-truth heatmap constructed using catch location and catch amount data (hereafter referred to as "catch data") and fishing vessel trajectory data (hereafter referred to as "trajectory data"). The following section describes the detailed methodology of this annotation process.

## Overview of our method

Fig 3 illustrates an overview of the proposed method. Our method consists of two phases: pre-training and fine-tuning. In the pre-training phase, the model is first trained with heatmaps generated from trajectory data. In the fine-tuning phase, training continues with heatmaps generated using catch data.

During the pre-training, we applied therefore meta-learning method proposed by Mostafa et al. [14] to control the contribution of noisy labels in the training data with trajectory data, which include unsuitable fishing locations (those with no catch or effort) or areas that do not correspond to the target species' habitat. In their method, network weights $w$ are updated as follows:

$$w_{t+1} = w_t - \frac{\eta_t}{b} \sum_{i=1}^{b} c_\theta\left(x_i, \tilde{y}_i\right) \nabla \mathcal{L}\left(f_{w_t}\left(x_i\right), \tilde{y}_i\right) \tag{1}$$

Pre-training with trajectory data

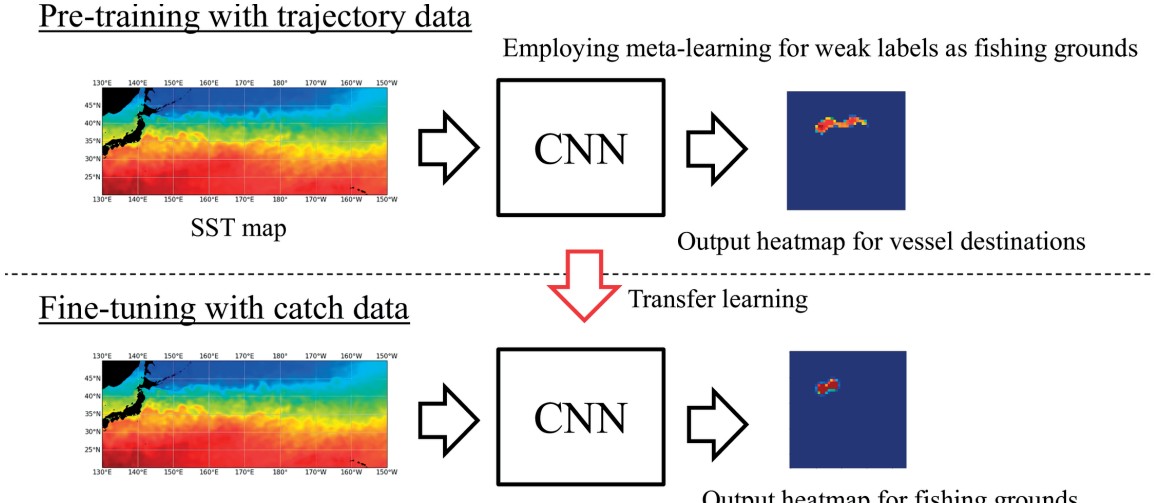

**Fig 3. Overview of the proposed method.** Our method involves pre-training with trajectory data as weak supervision and fine-tuning with catch data as strong supervision. In the pre-training phase, meta-learning is employed to reduce the effect of noisy labels.

where $c_\theta$ is the meta-learner named *confidence network*, which outputs the confidence for a pair of input instance $x_i$ and a weak label $\tilde{y}_i$. This learner calibrates the learning rate for each label. $f_{w_t}$ is the learner for the target task named *target network*. $\mathcal{L}$ is the loss function, $\eta_t$ is the global learning rate, and $b$ is the batch size. Since their method assumes a classification task and cannot be directly applied to our target task, we redefined the ground-truth confidence and the architecture of the *confidence network*. The Application of meta-learning section provides the details regarding the application method.

## Data

We use three types of data: SST, catch data, and fishing vessel trajectories.

SST is daily grid data based on satellite observations. An example of an SST map is shown in Fig. 4. Grids corresponding to land areas are treated as missing values and filled by zero. We use two-dimensional SST patterns as input because characteristic SST patterns, such as eddies and tides, can provide insights into estimating fishing grounds and suitable water temperatures for the target fish species.

Catch data based on fishing logbooks are tabular data that record fishing activity's operation date, geographic coordinates (latitude and longitude), and CPUE. Fig 4 shows an illustrative example where a red circle represents the location of a fishing ground. This data is a strong label derived directly from fishing vessels. In this study, we use this data to annotate ground-truth fishing grounds for the input images.

The fishing vessel trajectories are derived from the automatic identification system (AIS), which records an identification code along with historical data on the location of the fishing vessels, including date, location, and duration of stay. Fig 5 provides an illustrative example of this data. This study annotates these data, serving as weak labels with low-certainty. While the trajectories contain only information on the location of the fishing vessels and no information on the catch or location of the catch, it is possible to infer locations that are non-operational or likely to have been operational based on the speed of movement and movement patterns.

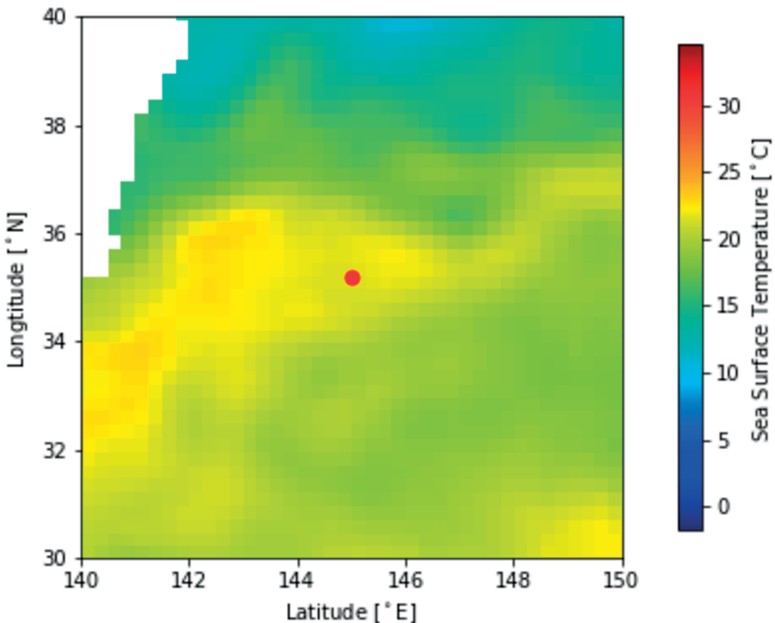

**Fig 4. Example of a sea surface temperature map and fishing ground (red circle).**

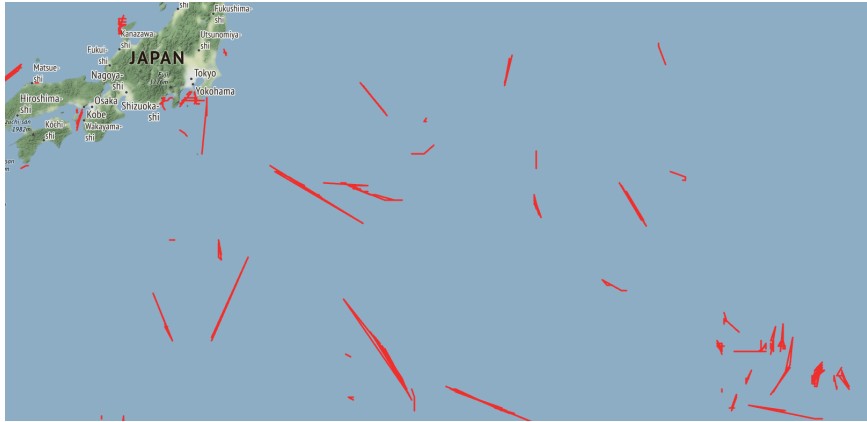

**Fig 5. Examples of fishing vessel trajectories.** A red line indicates the daily trajectory of each vessel.

For example, suppose a trajectory shown in Fig 5 is a straight line of a single day's travel distance length. In that case, no fishing operation likely took place along that path, and the existence of a fishing ground can be rejected. Conversely, if the daily distance of a trajectory is short, or if the trajectory shows repeated movement to and from the same point, it can be inferred with a high degree of certainty that fishing activity has occurred.

## Annotation

Training a keypoint-detection model requires ground-truth heatmaps representing the likelihood of the presence of keypoints. Our proposed approach generates heatmaps where the

fishing ground location has a high value, which gradually decreases as the distance from that location increases, as in heatmaps for human pose estimation.

As in the conventional methods, a two-dimensional Gaussian kernel in Eq (2) creates ground-truth heatmaps:

$$f(\boldsymbol{p}) = \exp\left(-\frac{\|\boldsymbol{p} - \boldsymbol{p}_{\mathrm{t}}\|_2^2}{2\sigma^2}\right) \tag{2}$$

where $\boldsymbol{p}$ represents the image coordinates, $\boldsymbol{p}_{\mathrm{t}}$ represents the ground-truth coordinates, and $\sigma$ represents the standard deviation.

To utilize positive examples (fishing ground class) and negative examples (non-fishing ground class) for model training, we define labels associated with the catch and trajectory data as shown in Table 1. *Speed* refers to the distance between the start and end points of a fishing vessel's daily activity. The labels "Good" and "Bad" indicate the quality of fishing grounds based on catch data, while the other labels are associated with trajectory data. "Unlikely" refers to a location with a significant daily travel distance where no fishing operation is estimated to have occurred. "Unknown" denotes a location where none of the above cases are present, and no distinction can be made between "Good" and "Bad" fishing grounds.

As discussed in the previous section, fishing vessel trajectories provide insights into potential fishing grounds, and locations unlikely to be fishing grounds. The catch data also provide information on locations where no catches were made or where catches were deemed unlikely, despite the occurrence of fishing operations.

In essence, the catch data and fishing trajectory data can identify three categories: (1) those that indicate instances where the fishery was operational and a certain quantity of fish were captured (good fishing grounds), (2) those that reflect instances where the fishery was operational or where it relocated to a particular location for operation but resulted in no fish being captured (bad fishing grounds), and (3) those that indicate instances where the fishery was never considered for operation initially (inappropriate fishing grounds). The proposed approach learns three types of heatmaps for each case to benefit from such negative examples. More precisely, the ground-truth heatmap comprises three channels corresponding to classes of fishing grounds. The classes are defined as "Good," "Bad," and "Unlikely," and the association with the label is determined as presented in Table 2. While only the "Good" channel evaluates, incorporating multi-task learning is expected to improve the generalization performance.

**Table 1. Definition of labels.**

| Label | Definition | Data |
|---|---|---|
| Good | $CPUE > 0$ | Catch |
| Bad | $CPUE = 0$ | Catch |
| Unlikely | $speed > 200$ km | Trajectory |
| Unknown | $speed \leq 200$ km | Trajectory |

**Table 2. Correspondence between labels and classes.**

| Class (Channel) | Label |
|---|---|
| Good | Good, Unknown |
| Bad | Bad, Unknown |
| Unlikely | Unlikely |

We prepare two types of ground-truth heatmaps: a heatmap using only catch data for fine-tuning and a heatmap using only trajectory data for pre-training, as shown in Fig 6. The locations with larger values in the heatmaps indicate a higher likelihood of the existence of fishing grounds.

## Keypoint detection model

For our fishing ground estimation, we employed Lightweight OpenPose [30], a streamlined version of OpenPose [29]. Note that our method is designed to be model-agnostic, so in principle it can be combined with any backbone architecture. Lightweight OpenPose was chosen for computational feasibility in our setting. The original OpenPose architecture outputs both keypoint heatmaps and part affinity fields to capture pairwise relations between keypoints. However, in our method, we focus exclusively on the keypoint heatmaps, which predict the likelihood of fishing grounds at each location on the SST map.

The model is composed of multiple convolutional layers that progressively refine the heatmaps across stages. Both the input SST maps and the output heatmaps are resized to maintain spatial consistency as Fig 7 shows. This approach leverages the efficiency of the lightweight architecture while focusing on keypoint detection for fishing ground estimation.

## Application of meta-learning

The work by Mostafa et al. [14] is designed for a classification task and is not directly applicable to the fishing ground estimation as a detection task. Our method redefines "confidence" and restores the *confidence network*.

In [14], a binary label that indicates whether a weak label is true represents the concept of ground-truth confidence. However, in our case, it is not feasible to categorize the accuracy of the weak label into a binary format, and the strong label y does not necessarily correspond to the truth. Consequently, we defined the degree of validity of the input-weak label pair as the confidence as follows:

$$c := 1 - l_{\mathrm{hm}} \tag{3}$$

$$l_{\mathrm{hm}} = \min\left(w_{\mathrm{count}} \sum_{i,j} \mathrm{ReLU}(y_{ij} - \tilde{y}_{ij}), 1\right) \tag{4}$$

$$\mathrm{ReLU}(x) = \max(0, x) \tag{5}$$

$$w_{\mathrm{count}} = 1/(n_{\mathrm{nonzero}} + 1) \tag{6}$$

where $\mathbf{y}$ denotes the heatmap of a two-dimensional array derived exclusively from the catch data, while $\tilde{\mathbf{y}}$ denotes the heatmap solely derived from the trajectory data. Through the operation of $\mathrm{ReLU}(\mathbf{y} - \tilde{\mathbf{y}})$, confidence diminishes when non-zero pixels in $\mathbf{y}$ coincide with zeros in $\tilde{\mathbf{y}}$, as seen in the bottom of Fig 9. This design is attributed to the enhanced reliability of $\mathbf{y}$ as a source of fishing ground locations. Conversely, confidence increases when pixels are zero in $\mathbf{y}$ but non-zero in $\tilde{\mathbf{y}}$ as indicated at the top of Fig 9. While regions with zero pixels in $\mathbf{y}$ are excluded from the confidence evaluation, this definition is grounded in the assumption that $\mathbf{y}$ excludes unobserved fishing grounds, thus making the region appear smaller than its actual extent. We adopted the same approach in the loss function introduced by [12]. $w_{\mathrm{count}}$ is the weighting factor to address the empirical observation where models trained only with catch data tend to generate smaller regions than the actual fishing grounds. This factor, proportional to the count of non-zero pixels in the weak label heatmap $\tilde{\mathbf{y}}$, augments the model's

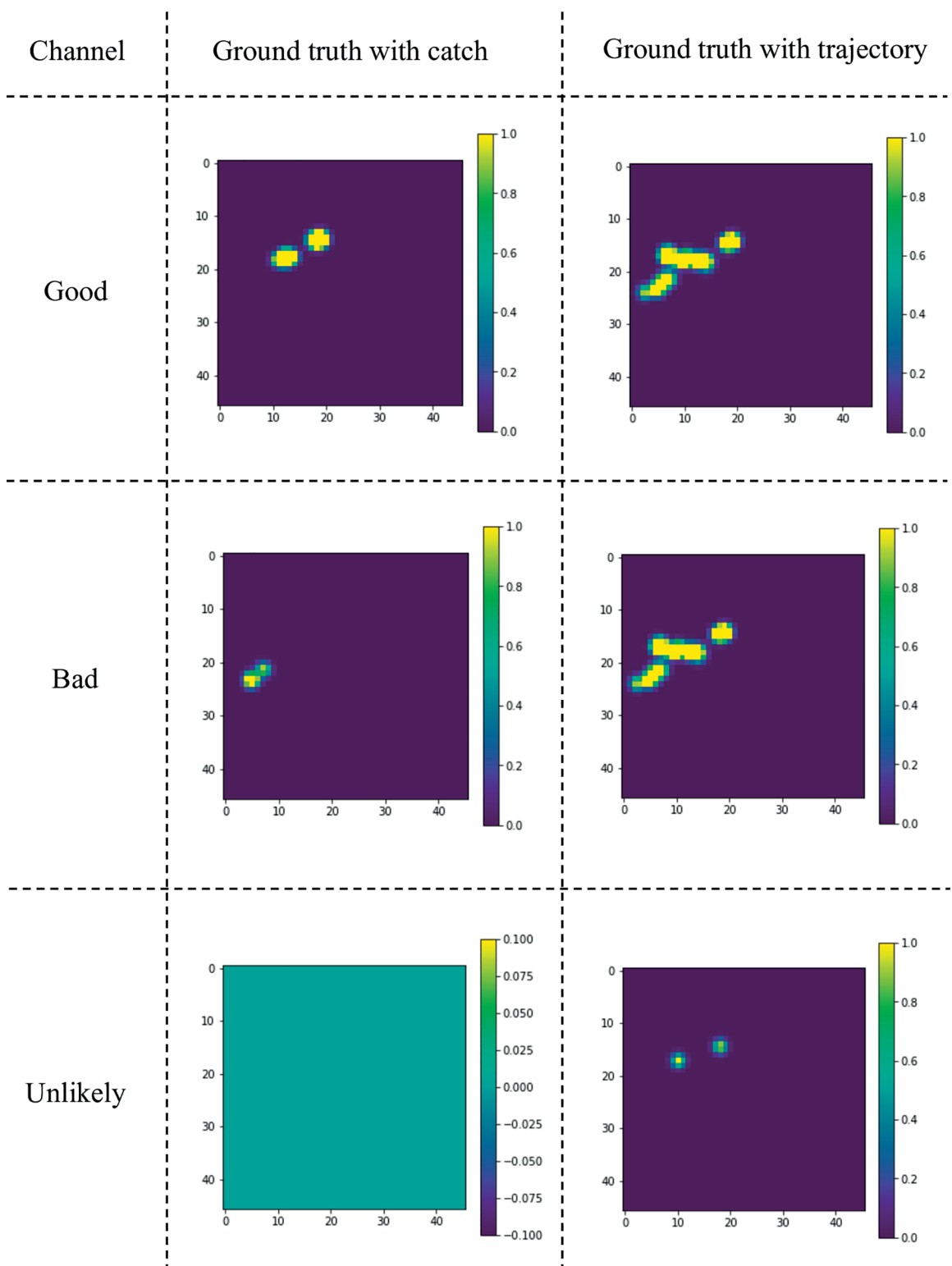

**Fig 6. Examples of ground-truth heatmaps.** The heatmaps with the trajectory data are the same for both "Good" and "Bad" channels. The "Unlikely" channel in the heatmap with the catch data all contains zero values.

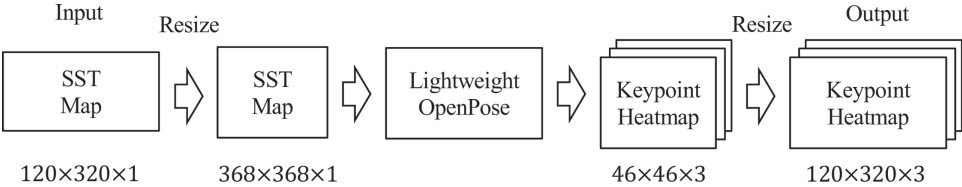

**Fig 7. Model input and output shapes.** The input and output image size is resized to fit the network architecture.

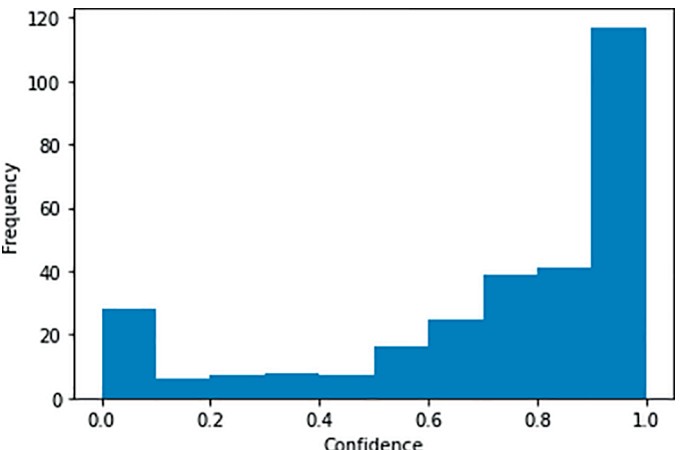

**Fig 8. Distribution of ground-truth confidence.**

confidence for samples characterized by a more extensive non-zero region. The objective is to motivate the model to prioritize learning from such samples, facilitating the generation of larger and more realistic fishing ground regions in its output.

Fig. 8 is the histogram of the confidence in the dataset we use, with approximately one-third of the samples having a confidence higher than 0.9.

The *confidence network* has an architecture as shown in Fig 10. The architecture is designed for two-class classification (with soft labels), using SST maps and ground-truth heatmaps generated from trajectory data as inputs. Each input image passes through two convolutional layers, and each feature map is flattened and concatenated. We used the convolutional layer with a kernel size of 3, zero padding, and a stride of 1. The activation function is the identity function for the output layer and ReLU for the other layers, while the sigmoid function is applied only to the output layer during inference.

## Evaluation

We evaluated the accuracy of the estimated fishing grounds by considering the geodesic distance, referred to as $d_{\text{diff}}$, between the peak coordinates derived from the output heatmap and the ground-truth coordinates. The evaluation criterion dictates that $d_{\text{diff}}$ should be less than 200 km, the estimated maximum distance a fishing vessel can move within a single day, assuming a speed of 14 knots (equivalent to 26 km/h) for approximately eight hours. Precision, recall, and F1-score are the performance metrics utilized in the evaluation process.

Ground truth with trajectory
(inference input for *confidence network*)

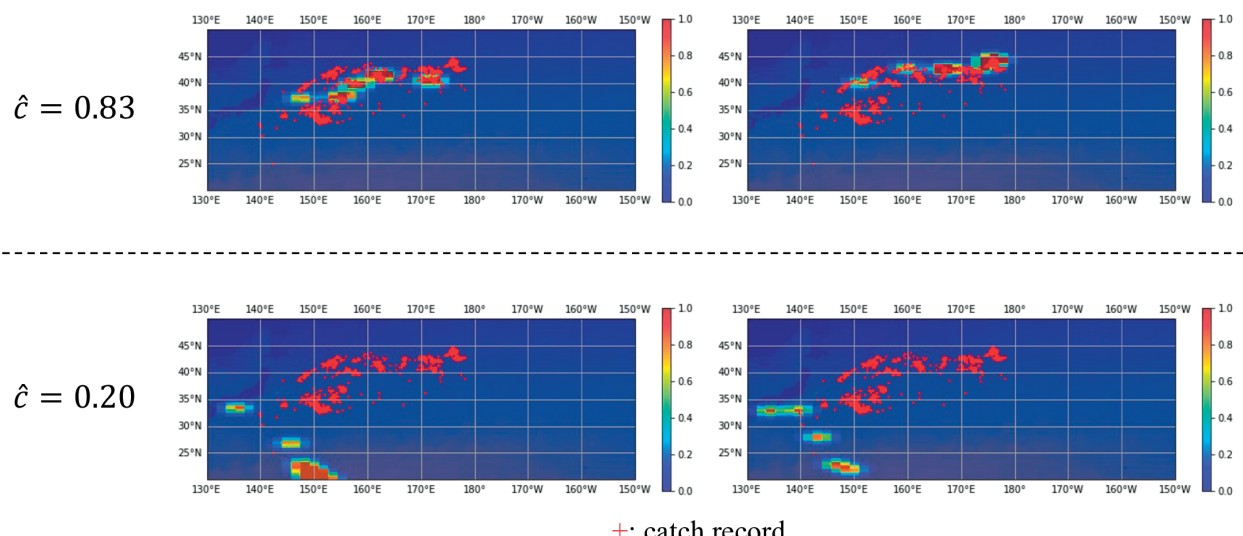

+: catch record

**Fig 9. Examples of correspondence between confidence and each ground-truth heatmap.**

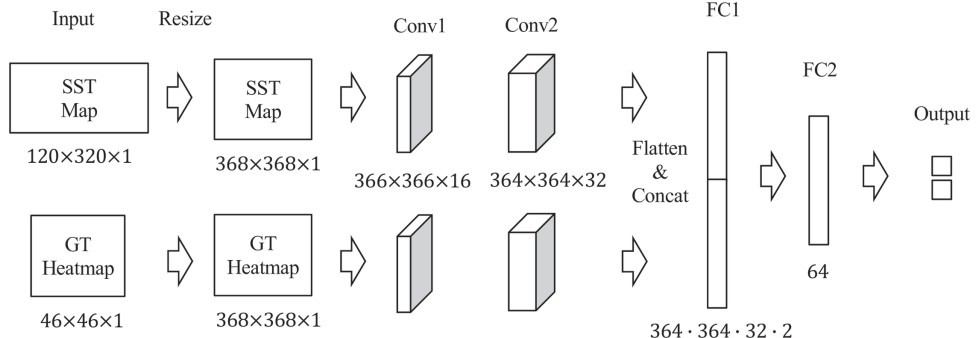

**Fig 10. Architecture of our *confidence network*.**

## Experiments

### Dataset

We obtained SST data from the National Oceanic and Atmospheric Administration (NOAA) [47] and catch data from the Shizuoka Prefectural Research Institute of Fishery and Ocean, specifically focusing on skipjack tuna fishing operation data. The fishing vessel trajectories were acquired from the Global Fishing Watch [22].

We set the designated sea area between 20°N and 50°N, between 130°E and 150°W, with a grid interval of 0.25°.

*Pre-processing.* The pre-processing of vessel trajectories involved selecting only single-line fishing vessels to ensure data consistency with our target species, skipjack tuna. Furthermore, we applied an SST filter, excluding any trajectory points where the SST was below 19°C or above 26°C, since skipjack tuna are typically found in these temperature ranges.

For generating ground-truth heatmaps, we applied the Gaussian kernel from Eq (2) with a standard deviation ($\sigma$) of 7 to either the catch location or the averaged coordinates of each vessel's daily activity to represent its location.

To prevent the over-densification of predicted points, peak detection was performed on the resulting heatmap using $8 \times 8$ maximum value filtering.

*Dataset composition.* We utilized 2087 days of trajectory data collected between 2012 and 2017 for pre-training the model. Additionally, 371 days of catch data from 2012 and 2015 were used for fine-tuning the model and training the *confidence network*. For testing, we employed 154 days of catch data from 2018. Table 3 summarizes the number of images and the instance counts for each label across the different phases of the experiment.

## Experimental design

The dataset was divided into training, validation, and test sets to ensure a robust evaluation of model performance. Trajectory data from 2012 to 2017 was utilized for pre-training the target network, while catch data from 2012 to 2015 was split into training and validation sets, with a 0.2 validation split ratio applied. This catch data was also used to fine-tune the target network and to train the confidence network. The test set, consisting of 154 days of catch data from 2018, was reserved exclusively for final evaluation.

*Training and hyperparameter selection.* The training process involved two main stages: pre-training and fine-tuning. During pre-training, the model learned from trajectory data, and fine-tuning adjusted the network using catch data. In addition, the confidence network was trained on catch data, and its inference output was used as a weight to guide the fine-tuning of the target network.

Table 4 shows the hyperparameters used in the training for each network. The epoch of training for the *target network* was determined by the highest validation F1-score and the one for *confidence network* was determined by the lowest validation loss. The other hyperparameters were tuned empirically, balancing convergence speed and stability.

## Results

We compared our proposed method (*w/confidence*) with the following three patterns: (*catch only*) trained using only catch data, (*fine-tuning*) our previous method of pre-training with heatmaps using both trajectories and catch data [13] and (*w/o confidence*) our method without confidence in pre-training, which corresponds to existing weakly supervised learning methods.

Table 5 shows the experimental results. For the F1-score and recall, *w/confidence* achieved the highest performance. The precision for *catch only* was the highest, but the number of predictions and the recall were the lowest, indicating an under-detection problem. The methods using trajectory data for training increased the number of predictions and improved recall, and *w/o confidence* and *w/confidence* showed further improvement because about five times more days were used for pre-training.

**Table 3. The Number of images and instances for each label in the dataset. "-" denotes a non-existent item.**

|  | Images | Good | Bad | Unlikely | Unknown |
|---|---|---|---|---|---|
| Pre-training (trajectory) | 2087 | - | - | 2487 | 14196 |
| Fine-tuning (catch) | 371 | 2942 | 1480 | - | - |
| Test (catch) | 154 | 1606 | - | - | - |

**Table 4. Hyperparameter setting.**

|  | confidence network | target network |
|---|---|---|
| Loss | Cross entropy | L2 |
| Optimizer | Adam | Adam |
| Learning rate | 1e-5 | 4e-5 |
| Batch size | 16 | 16 |
|  |  | Pre-training: 100 |
| Epoch | 19 | Fine-tuning: 33 |

Fig 11 illustrates a visualization of output heatmaps. Other trajectory data methods identify fishing locations not covered by the *catch only* approach. Although the outputs of the three methods with trajectory data are similar, there are cases where the *w/ confidence* approach produces more points or provides more accurate locations.

## Discussion

### Effectiveness of meta-learning

Fig 13 shows examples of the ground-truth heatmaps for the high and low values generated by the *confidence network* during pre-training. In the case of high confidence, the heatmap exhibits larger non-zero areas or suitable areas for fishing grounds (similar to the spatial distribution of the catch record). Conversely, low-confidence instances typically feature heatmaps with smaller non-zero areas or areas unsuitable for identifying fishing grounds, which are rarely observed in the catch record. Fig 12 shows the distribution of confidence values during pre-training. As there is a larger number of low-confidence samples, the latter type of heatmap has a diminished impact on model weight updates, contributing to improved performance during fine-tuning. Table 6 presents the results of evaluating the pre-trained model on the test data. *w/ confidence* aligns well with the catch data, indicating that the improved similarity between source and target tasks in transfer learning leads to enhanced fine-tuning performance.

### Change of loss function

We employed a modified loss function to address the under-detection instead of using trajectory data, and we examined the synergistic effects. Hinge loss proposed in [12] is defined as

**Table 5. Evaluation results.**

| Training method | Precision | Recall | F1-score | # of peaks |
|---|---|---|---|---|
| *catch only* | **0.233** | 0.031 | 0.055 | 215 |
| *fine-tuning* [13] | <u>0.215</u> | 0.043 | 0.072 | 330 |
| *w/o confidence* | 0.181 | <u>0.054</u> | <u>0.083</u> | <u>475</u> |
| *w/confidence* | 0.196 | **0.059** | **0.090** | **480** |

bold: best score
underline: second best score

**Table 6. Evaluation results for pre-training.**

| Pre-training | Precision | Recall | F1-score | # of peaks |
|---|---|---|---|---|
| *w/o confidence* | 0.065 | 0.006 | 0.011 | 155 |
| *w/ confidence* | 0.097 | 0.026 | 0.040 | 421 |

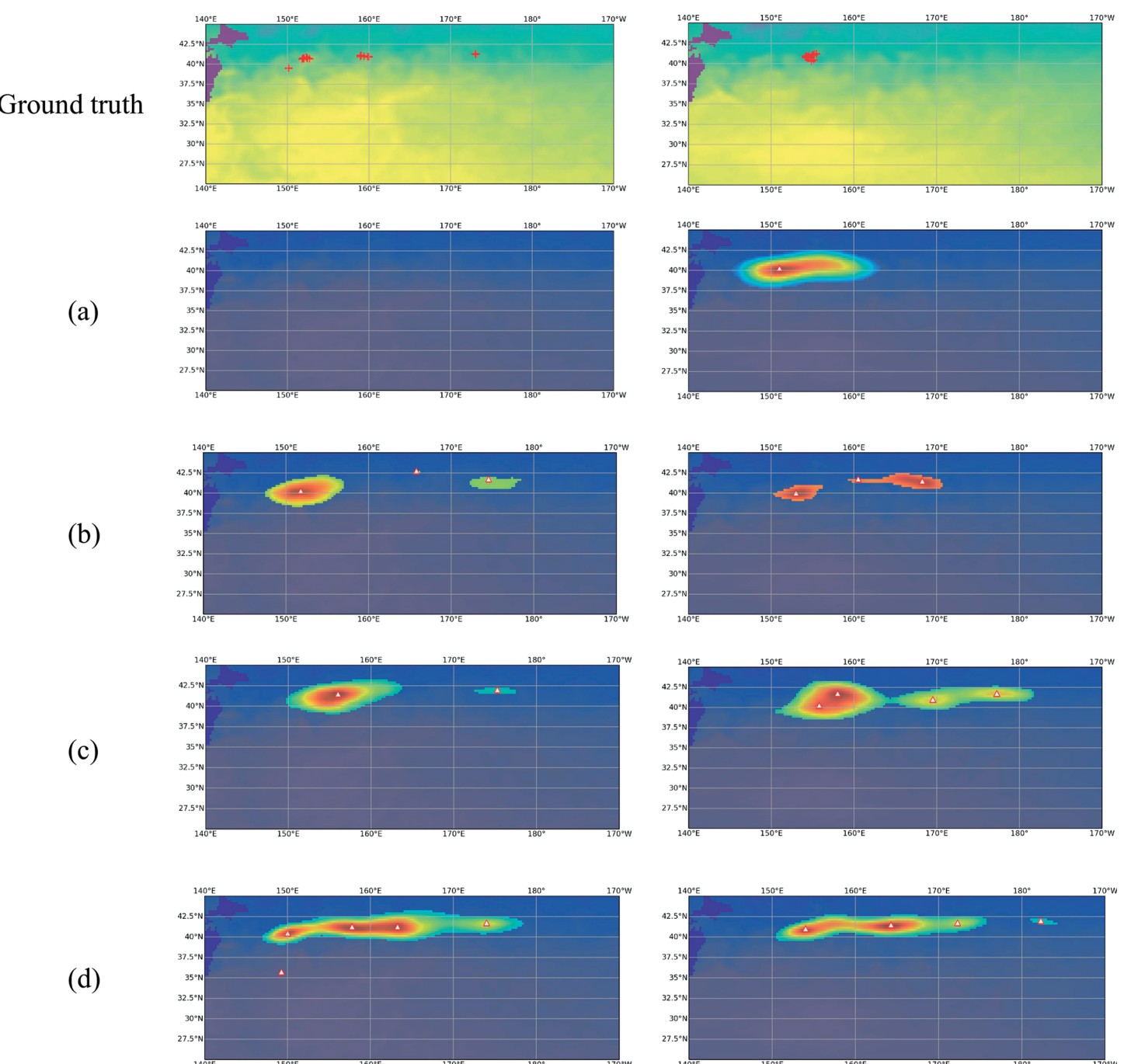

**Fig 11. Examples of output heatmaps overlapped with SST maps**: (a) *catch only* (b) *fine-tuning* (c) *w/o confidence* (d) *w/ confidence*. △ represents the estimated point and + represents the ground-truth coordinate.

follows:

$$L_2 = \frac{1}{2} \sum_c \sum_{i,j} (\hat{y}_{cij} - y_{cij})^2 \tag{7}$$

$$L_H = \sum_c \sum_{i,j} \text{ReLU}(y_{cij} - \hat{y}_{cij}) \tag{8}$$

$$L = \alpha L_2 + (1 - \alpha) L_H \tag{9}$$

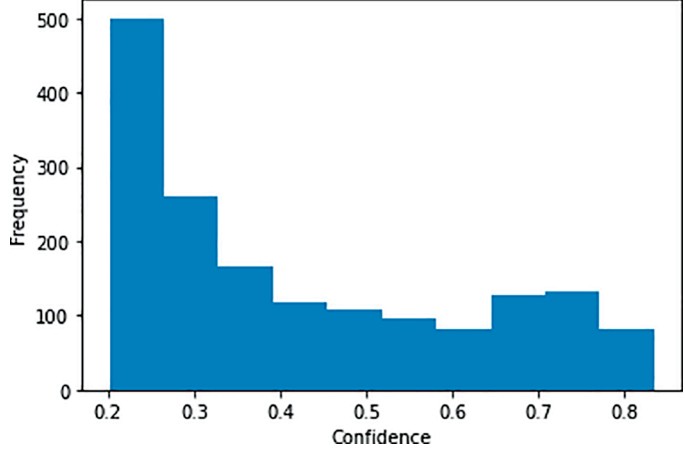

**Fig 12. Distribution of confidence in pre-training.**

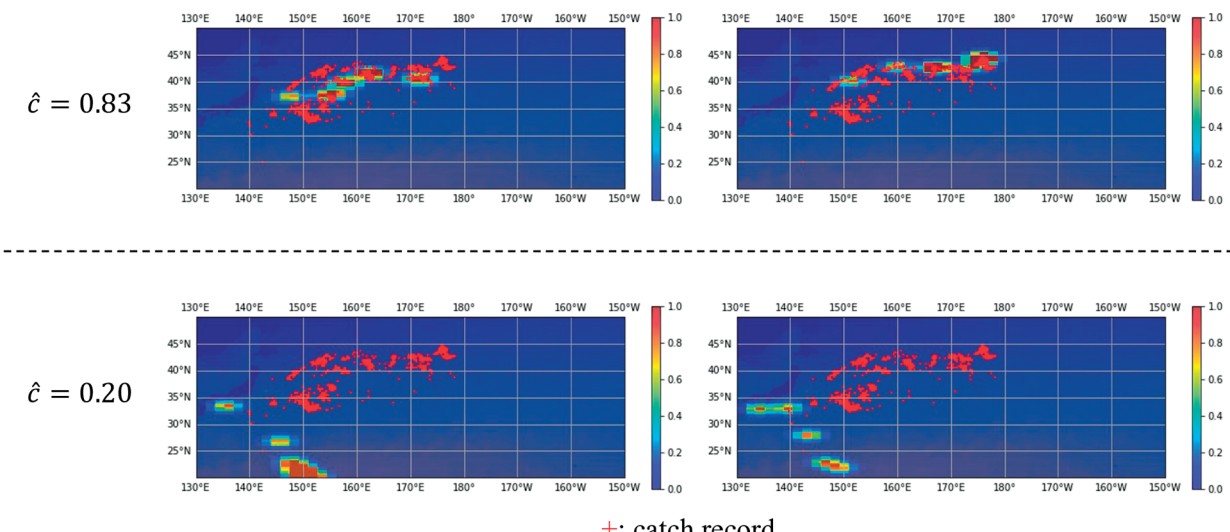

$+$: catch record

**Fig 13. Examples of input heatmaps $\tilde{y}$ in the cases where the confidence network output values are high and low.** They are resized for visualization. A red marker represents a catch record location in the training data (2012).

where $c$ is a channel of a ground-truth heatmap, and $\alpha$ is a hyperparameter between 0 and 1. $L_{\mathrm{H}}$ sets the over-detection loss to zero, leading to an increase in the number of predictions.

However, as shown in Table 7, the experimental results of using this loss function revealed that while the F1-score improved for the *catch only*, it did not improve for the *w/ confidence*. This discrepancy can be explained by the fact that hinge loss does not penalize over-detection effectively. Specifically, hinge loss calculates the penalty for over-detection as 0 regardless of the output position, which leads to a situation where, in the pre-trained state where the number of predictions has already increased, additional predictions are made for unsuitable fishing locations. These predictions do not represent accurate fishing grounds, causing a decrease in precision without a corresponding increase in recall.

This behavior suggests that hinge loss, while helpful in increasing predictions to avoid under-detection, may not align with the specific objectives of our task. In particular, the

**Table 7. Evaluation results for changes in the loss function.**

| Training Method | Loss Function | Precision | Recall | F1-score | # of peaks |
|---|---|---|---|---|---|
| *catch only* | L2 | **0.233** | 0.031 | 0.055 | 215 |
| *catch only* | Hinge ($\alpha = 0.5$) | 0.164 | 0.036 | 0.059 | 353 |
| *catch only* | Hinge ($\alpha = 0.1$) | 0.162 | 0.047 | 0.073 | 468 |
| *w/ confidence* | L2 | <u>0.196</u> | <u>0.059</u> | **0.090** | 480 |
| *w/ confidence* | Hinge ($\alpha = 0.5$) | 0.144 | 0.058 | <u>0.083</u> | <u>646</u> |
| *w/ confidence* | Hinge ($\alpha = 0.1$) | 0.104 | **0.067** | 0.082 | **1091** |

bold: best score
underline: second best score

balance between increasing predictions and maintaining precision becomes critical. As the model starts to predict more locations, including unsuitable ones, the lack of penalty for over-detection leads to a decrease in precision without a beneficial increase in recall. Therefore, the use of hinge loss in this context requires further optimization to ensure that the increase in predictions does not degrade model performance in terms of precision.

## Limitations

The F1-scores from our experiments are significantly lower than those of general keypoint-detection tasks such as human pose estimation. This performance gap can be attributed to the intrinsic complexity of the task. Unlike human keypoints, which follow anatomically fixed positions, fishing grounds are dynamic and influenced by oceanographic conditions, vessel behavior, and ecological factors. Additionally, while human pose estimation benefits from precise, manually annotated datasets, fishing ground labels are inferred from incomplete catch data and noisy trajectory data, introducing additional uncertainty.

The incompleteness of catch data annotations hindered our model's performance, affecting both fine-tuning and meta-learning strategies relying on accurate ground-truth heatmaps derived from catch data. A potential strategy to improve fishing ground annotations is incorporating sonar data from buoys and vessels. Brehmer et al. demonstrated omnidirectional multibeam sonar, commonly found on fishing vessels, provides real-time monitoring of fish schools, offering valuable data for more accurate ground identification. Integrating sonar data with catch and trajectory data can increase the volume of reliable data, improving the overall quality of annotations.

Additionally, the precision of our method faces limitations in surpassing the precision achieved through training solely with catch data, emphasizing the importance of ensuring high precision in *catch only*. To achieve higher precision, additional input features, such as SSH and sea current velocity, which play a role in shaping fishing grounds, should be included. Moreover, refining the network architecture to better capture spatial dependencies remains an essential direction for future work. The resolution of the heatmaps also plays a crucial role in the performance, as fishing grounds are not fixed points but spatially distributed regions. Higher-resolution heatmaps could allow for finer localization of fishing grounds.

We evaluated the distance using the geodetic distance threshold (200 km). This was determined based on the distance that fishing vessels can travel in a day and the resolution of the SST data, but if higher resolution water temperature data can be obtained, it should be possible to evaluate it over shorter distances.

Lastly, the experimental situations, including the target fish species, target areas, and target years, are currently limited. While this study focuses on skipjack tuna as a representative

case, the proposed method is not species-specific and can be extended to other fisheries with appropriate data. Future research should explore broader applications to different species and regions to further assess the generalization of the approach.

## Conclusion

We proposed a method for estimating fishing grounds, which involves meta-learning for initial pre-training using fishing vessel trajectories as a form of weak supervision, followed by fine-tuning with a limited quantity of catch data. Our experimental results confirmed the effectiveness of the pre-training and meta-learning methods.

Our future work will address the identified limitations to enhance the proposed method. Specifically, we aim to augment the catch dataset by incorporating accurate sonar data from buoys and vessels to improve the accuracy of annotations. Additionally, we will enhance the precision of our method by including additional input features such as SSH and sea current velocity, optimizing the network architecture for better performance. Furthermore, we plan to evaluate the versatility of the proposed method in a more diverse range of settings, including different target fish species, areas, and years, to ensure its applicability across various scenarios.

## Acknowledgments

We sincerely thank Shizuoka Prefectural Research Institute of Fishery and Ocean for providing the fish catch data essential to our research.

## Author contributions

**Conceptualization:** Masaaki Iiyama.

**Data curation:** Masaaki Iiyama, Kazuki Takasan.

**Funding acquisition:** Masaaki Iiyama.

**Methodology:** Masaaki Iiyama.

**Project administration:** Masaaki Iiyama.

**Software:** Kazuki Takasan.

**Supervision:** Masaaki Iiyama.

**Validation:** Kazuki Takasan.

**Visualization:** Kazuki Takasan.

**Writing – original draft:** Kazuki Takasan.

**Writing – review & editing:** Masaaki Iiyama.

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
