## [Decision Letter · Decision Letter 0]

20 Sep 2024

PONE-D-24-20285Fishing ground estimation through weakly supervised keypoint detection and meta-learningPLOS ONE

Dear Dr. Iiyama,

Thank you for submitting your manuscript to PLOS ONE. After careful consideration, we feel that it has merit but does not fully meet PLOS ONE’s publication criteria as it currently stands. Therefore, we invite you to submit a revised version of the manuscript that addresses the points raised during the review process.

We look forward to receiving your revised manuscript.

Kind regards,

Lei Chu

Academic Editor

PLOS ONE

“Japan Society for Promotion of Science,  KAKENHI, Grant Number 21H04913.

Japan Science and Technology Agency, CREST, Grant Number JPMJCR19F1.”

Additional Editor Comments:

After careful evaluation, we find the manuscript promising but believe that minor revisions are necessary to improve clarity and detail. Specifically, we recommend restructuring the introduction into distinct paragraphs to better emphasize the research background, limitations of existing methods, and contributions of the paper. Additionally, please provide more concise descriptions of keypoint detection in the method section, expand on the data preprocessing and annotation processes, and add detailed information about the experimental design, including dataset division, hyperparameters, and evaluation metrics.

Reviewers' comments:

Reviewer's Responses to Questions

**Comments to the Author**

1. Is the manuscript technically sound, and do the data support the conclusions?

Reviewer #1: Yes

2. Has the statistical analysis been performed appropriately and rigorously? 

Reviewer #1: Yes

3. Have the authors made all data underlying the findings in their manuscript fully available?

Reviewer #1: Yes

4. Is the manuscript presented in an intelligible fashion and written in standard English?

Reviewer #1: No

5. Review Comments to the Author

Reviewer #1: This study introduces a method to estimate fishing grounds using weakly supervised keypoint detection and meta-learning. Traditional methods, such as habitat suitability index models and statistical approaches, often overlook broader environmental patterns crucial for fishers. This research focuses on leveraging sea surface temperature (SST) patterns to predict fishing ground locations, viewing the fisher's decision-making process as a pattern recognition task. However, there are areas for improvement regarding the readability of the paper and the scientific contribution as follows:

1. The introduction contains a lot of information but lacks a clear paragraph structure. Please divide the introduction into several paragraphs, each focusing on a specific topic, such as research background, limitations of existing methods, and contributions of this paper.

2. When describing the method, there is a lack of concise description of the specific details of keypoint detection.

3. The description of the data is not detailed enough, especially the process of data preprocessing and annotation generation.

4. The description of the experimental design and the division of the data set is not detailed enough. Please add detailed description of the experimental design, including the division of the data set (training set, validation set and test set), the selection of hyperparameters, evaluation indicators, etc.

6. PLOS authors have the option to publish the peer review history of their article (what does this mean?). If published, this will include your full peer review and any attached files.

Reviewer #1: No

---

## [Author Response · Author response to Decision Letter 1]

15 Oct 2024

We sincerely thank the reviewers for carefully reading our manuscript and giving us constructive comments to improve our paper. We have made changes that reflect most of the suggestions we received from the reviewers. We have highlighted the changes in the manuscript.

Point-by-point response to the reviewers' comments is written in "Response to Reviewers.pdf". We hope that our revisions meet with your approval and adequately address all concerns raised.

---

## [Decision Letter · Decision Letter 1]

30 Dec 2024

PONE-D-24-20285R1Fishing ground estimation through weakly supervised keypoint detection and meta-learningPLOS ONE

Dear Dr. Iiyama,

Thank you for submitting your manuscript to PLOS ONE. After careful consideration, we feel that it has merit but does not fully meet PLOS ONE’s publication criteria as it currently stands. Therefore, we invite you to submit a revised version of the manuscript that addresses the points raised during the review process.

Key challenges include the reliance on fishing data, which limits generalizability, and low F1-scores compared to traditional tasks, highlighting the need for deeper analysis and stricter evaluation metrics. The use of noisy trajectory data for weak supervision requires stronger validation, while the innovative application of meta-learning lacks comparisons to simpler alternatives like data augmentation. Expanding beyond skipjack tuna to multiple species and including comparisons with state-of-the-art methods in weakly supervised learning could enhance the study's relevance and impact. Furthermore, exploring alternative model architectures, improving the clarity of visualizations, and re-evaluating the suitability of hinge loss would provide deeper insights and strengthen the findings. Integrating environmental data (e.g., sea currents) could also enrich the dataset and improve accuracy.

We look forward to receiving your revised manuscript.

Kind regards,

Lei Chu

Academic Editor

PLOS ONE

Journal Requirements:

Additional Editor Comments (if provided):

Reviewers' comments:

Reviewer's Responses to Questions

**Comments to the Author**

1. If the authors have adequately addressed your comments raised in a previous round of review and you feel that this manuscript is now acceptable for publication, you may indicate that here to bypass the “Comments to the Author” section, enter your conflict of interest statement in the “Confidential to Editor” section, and submit your "Accept" recommendation.

Reviewer #2: (No Response)

Reviewer #3: All comments have been addressed

2. Is the manuscript technically sound, and do the data support the conclusions?

Reviewer #2: Partly

Reviewer #3: Yes

3. Has the statistical analysis been performed appropriately and rigorously? 

Reviewer #2: Yes

Reviewer #3: Yes

4. Have the authors made all data underlying the findings in their manuscript fully available?

Reviewer #2: Yes

Reviewer #3: Yes

5. Is the manuscript presented in an intelligible fashion and written in standard English?

Reviewer #2: Yes

Reviewer #3: Yes

6. Review Comments to the Author

Reviewer #2: The paper titled "Fishing ground estimation through weakly supervised keypoint detection and meta-learning" explores innovative methods for estimating fishing grounds, which are crucial for optimizing fishing activities. The authors, Masaaki Iiyama and Kazuki Takasan, focus on the decision-making processes of fishers, particularly how they utilize sea surface temperature (SST) patterns to identify potential fishing areas. The study recognizes the limitations of traditional empirical methods that rely on environmental data, which often yield inconsistent results due to the complexities of marine environments and variations in fisher expertise. To address the challenge of insufficient annotated data for training predictive models, the authors propose a novel framework that combines weakly supervised pre-training with meta-learning. They leverage publicly available trajectory data from fishing vessels as a form of weak supervision. While this trajectory data does not always provide precise locations of fishing grounds, it offers valuable insights into broader patterns of fishing activity. During the pre-training phase, the model learns these patterns from trajectory data, which enhances its ability to recognize SST characteristics relevant to fishing.

The authors emphasize the importance of fine-tuning the model with more accurate catch data after the initial pre-training. This dual-phase approach aims to improve prediction accuracy despite the scarcity of reliable catch data. Additionally, to counteract the impact of noisy labels present in trajectory data—such as locations where vessels did not actually fish—the authors introduce a meta-learner. This component evaluates label reliability and minimizes the influence of misleading data on the model's learning process. The experimental results presented in the paper demonstrate that their proposed methods significantly enhance the estimation of fishing grounds compared to traditional approaches. The findings underscore the effectiveness of integrating weakly supervised learning and meta-learning techniques in addressing data limitations in marine resource management. Overall, this research contributes valuable insights into utilizing advanced machine learning methodologies for practical applications in the fishing industry, ultimately aiming to improve efficiency and sustainability in fishing practices.

Authors might consider the following comments:

• The title emphasizes the methodology ("weakly supervised keypoint detection and meta-learning") but does not highlight the outcome or potential implications (e.g., enhanced fishing efficiency, improved ecological management). Therefore, I would suggest to shift the focus toward the practical impact of the research. For example: "Improving Fishing Ground Predictions with Weak Supervision and Meta-Learning".

• While the abstract mentions "effectiveness of pre-training and meta-learning," it does not provide quantitative outcomes or a concrete comparison to existing methods. Including key performance metrics (e.g., improvement in F1-score) would better highlight the study's contributions. Additionally, terms like "weakly supervised pre-training" and "meta-learning" may be unclear to readers unfamiliar with machine learning. A brief simplification or explanation of these terms would improve accessibility.

• In my opinion, while the introduction highlights the scarcity of annotated data as a motivation for weakly supervised learning, it lacks a broader context about how this approach compares to or complements other learning paradigms in similar domains (e.g., transfer learning, semi-supervised learning). Also, the focus on pattern recognition in sea surface temperature (SST) data could be enriched by drawing analogies to neural mechanisms involved in processing spatiotemporal patterns, such as those in the visual cortex. In this regards, authors might consider further discussing how the decision-making process in fishers parallels neural mechanisms in humans, such as how the prefrontal cortex integrates environmental cues (e.g., SST patterns) to make predictions and decisions [https://doi.org/10.3390/ijms25052724; https://doi.org/10.1016/j.brat.2024.104548;
https://doi.org/10.1111/nyas.15145].

• The reliance on fishing vessel trajectory and catch data limits the scope of generalizability. Additional sources, such as environmental factors (e.g., sea current velocity, chlorophyll levels), could enrich the dataset and enhance model accuracy.

• The F1-scores for the proposed method are significantly low, especially compared to traditional keypoint detection tasks. This discrepancy needs a deeper discussion of potential causes and strategies for improvement beyond the suggestions in the limitations section.

• The geodesic distance threshold (200 km) for evaluating accuracy might be overly lenient. A stricter metric could provide better insights into the method's precision and applicability.

• The paper highlights noisy and imprecise trajectory data influencing pre-training. The justification for using trajectory data as weak supervision requires further validation to strengthen the rationale for this approach.

• The use of meta-learning to mitigate noisy labels is innovative, but the paper lacks quantitative comparisons against simpler alternatives (e.g., data augmentation or filtering). Demonstrating the necessity and added value of meta-learning in this context would solidify the contribution.

• While the authors compare their method to alternatives like "catch only" and "fine-tuning," the addition of comparisons to state-of-the-art methods for weakly supervised learning in similar domains could enhance the impact of the results.

• The choice of Lightweight OpenPose for keypoint detection is reasonable, but an ablation study exploring the impact of alternative architectures (e.g., High-Resolution Networks or DeepLab) could provide more depth to the findings.

• The experimental focus on skipjack tuna limits the study's broader applicability. Expanding the analysis to include multiple species or regions could make the method more universally relevant.

• Figures showing heatmaps and output predictions (e.g., Fig. 11) could benefit from enhanced clarity and consistent formatting. Overlapping visualizations could obscure key findings, so better differentiation between estimated and ground-truth points would help.

• The results using hinge loss indicate decreased precision in some cases. This requires a more thorough examination and explanation of why hinge loss might not align with the task's objectives.

Reviewer #3: A technically sound study and displayed a rigour and depth perspectives of application of OpenPose & its experimental design

7. PLOS authors have the option to publish the peer review history of their article (what does this mean?). If published, this will include your full peer review and any attached files.

Reviewer #2: No

Reviewer #3: No

---

## [Author Response · Author response to Decision Letter 2]

27 Feb 2025

We appreciate the detailed feedback provided in the decision letter. We have prepared a separate document, "Response to Reviewers.pdf", that contains comprehensive, point-by-point responses to all the reviewer and editor comments. Kindly refer to the attached file for a full account of our revisions and clarifications.

---

## [Editor Report · Decision Letter 2]

2 Mar 2025

Improving Fishing Ground Estimation with Weak Supervision and Meta-Learning

PONE-D-24-20285R2

Dear Dr. Iiyama,

We’re pleased to inform you that your manuscript has been judged scientifically suitable for publication and will be formally accepted for publication once it meets all outstanding technical requirements.

Kind regards,

Lei Chu

Academic Editor

PLOS ONE
---

## [Editor Report · Acceptance letter]

PONE-D-24-20285R2

PLOS ONE

Dear Dr. Iiyama,

I'm pleased to inform you that your manuscript has been deemed suitable for publication in PLOS ONE. Congratulations! Your manuscript is now being handed over to our production team.

Kind regards,

on behalf of

Dr. Lei Chu

Academic Editor

PLOS ONE